# 24-hour movement behaviors and cognitive ability in preschool children: A compositional and isotemporal reallocation analysis

Zhihan Xu[1,2], Shiqiang Wang[1,2]*, Zitong Ma[3], Dan Li[1], Shuge Zhang[1,2,4]

1 College of Physical Education, Hunan University of Technology, Zhuzhou, Hunan, China, 2 Hunan Research Centre for Excellence in Fitness, Health and Performance, Zhuzhou, China, 3 School of Educational Science, Hunan Normal University, Changsha, China, 4 School of Human Sciences, University of Derby, United Kingdom

* suswsq@163.com

## Abstract

Research has supported the association between movement behaviors and cognitive ability in preschool children. However, most of the research has independently examined the various movement behaviors (e.g., physical activity, sedentary behavior, and sleep) without considering the dynamic composition of these behaviors in a 24-hour daily cycle. Therefore, this study aimed to explore the relationship between 24-hour movement behaviors and cognitive ability in preschool children. The participants were 191 Chinese preschool children from Zhuzhou aged 3–6 years. We measured light physical activity (LPA), moderate-to-vigorous physical activity (MVPA), and sedentary behavior (SB) using the Actigraph Accelerometers and evaluated the children's sleep time based on reports from parents and teachers. The Chinese version of the Wechsler Young Children Scale of Intelligence (C-WYCSI) was used to assess cognitive ability. Compositional analysis and isotemporal substitution were performed to examine the influence of 24-hour movement behaviors on children's cognitive ability. After controlling for demographics (e.g., age and sex), the composition of 24-hour movement behaviors was significantly associated with the verbal intelligence quotient (VIQ), performance intelligence quotient (PIQ), and full intelligence quotient (FIQ). Importantly, preschool children demonstrated greater cognitive behavior when time in MVPA replaced that spent in LPA, SB, or sleep than when time spent in LPA, SB, and sleep replaced that spent in MVPA within the 24-hour cycle. Physical activity engaged in a 24-hour daily cycle has a significant effect on cognitive ability in preschool children, with increased MVPA and LPA being associated with higher PIQ and FIQ and increased sleep being associated with lower VIQ, PIQ, and FIQ. Replacing time in SB and LPA with MVPA is promising for children's cognitive development.

## Introduction

Cognitive ability refers to the psychological processes of acquiring and understanding knowledge through thinking, experience, and sensory perception, such as executive function (EF), language, intelligence, attention and perception [1]. Children are accompanied by

**Data availability statement:** All relevant data are within the paper and its Supporting Information files.

**Funding:** Hunan Provincial Education Science Planning Project Base Project, grant number XJK23AJD055. the funder of this study, Shiqiang Wang, had been awarded the Hunan Provincial Education Science Planning Project Base Project (grant number: XJK23AJD055). Investors play an important role in article quality monitoring, publication decision-making, review, and editing. Finally, we confirm that there is no conflict of interest in this study.

**Competing interests:** The authors have declared that no competing interests exist.

the development of cognitive ability from birth[2]. Early childhood is a sensitive period for cognitive development, during which healthy brain development promotes optimal cognitive development and lays the foundation for future cognitive and academic achievements [3]. In addition, studies have shown that cognition is an essential indicator for measuring the development of an individual's nervous system and brain and can significantly predict cognitive ability in old age [4]. Therefore, the development of cognitive ability during the preschool period is crucial for an individual's health.

A reasonable and healthy lifestyle, such as adequate physical activity (PA) [5], less sedentary behavior (SB) [6], and high-quality sleep[7], is vital to the cognitive development of preschool children. Previous studies have shown that PA promotes memory-related cognitive development by mediating the expression of brain-derived neurotrophic factor (BDNF) and neurogenesis in the hippocampus [8]. In contrast, excessive SB is associated with lower white matter in the brain and has a negative impact on cognitive ability in young children [9]. Sleep improves neural plasticity through physiological mechanisms such as synaptic remodeling, thereby promoting cognitive development in young children [10]. Although existing systematic reviews concluded that PA are associated with cognitive ability in preschool children [11]. Due to differences in PA protocols and results reported in previous research reports, it is unclear which elements of PA are more beneficial for cognitive abilities in preschool children. Many studies emphasize the important role of MVPA in the cognitive development of preschool children [12, 13]. Quan et al. [14] reported that MVPA has no significant association with cognitive ability in preschool children, and that only boys with LPA promote cognitive development. Moreover, different sleep parameters also affect the development of cognitive ability in young children [15]. Peled et al. [16] found that sleep duration is not related to working memory in preschool children, while sleep disorders are associated with poor performance in working memory. Zhou et al. [7] found that total sleep duration is related to cognitive development in preschool children. Among them, moderate duration sleep has better benefits for cognitive ability than prolonged sleep, while insufficient sleep has the greatest harm to cognitive development. Similarly, the association between SB and cognitive development in young children should also vary depending on the type of SB [17]. Hutton et al. [9] found a negative relationship between screen-based SB and cognitive abilities in preschool children. However, Li et al. [18] suggests that not all types of SB are harmful, and non-screen SB has a positive impact on FIQ in preschool children. In addition, the study also found that prolonged sitting is beneficial for cognitive development in preschool children.

Although current research has focused mainly on analyzing the relationship between single-movement behavior and health from a "fragmented" perspective, this approach is not without limitations. Previous studies have shown that PA and sleep may jointly affect cognitive ability [19], suggesting that Adequate PA may alleviate some adverse effects of sleep deprivation on cognition, and sufficient sleep may be a factor in enhancing cognition through PA. Recent research suggests that a more systematic and comprehensive approach is needed to analyze the relationship between daily movement behavior and cognitive development in young children [20]. In fact, PA, SB, and sleep make up 24 hours a day. A change in the time of one movement behavior inevitably leads to a change in the time of one or more other movement behaviors, and each movement behavior is interdependent and mutually influential [21]. Therefore, all movement behaviors should be viewed as a compositional whole.

There are cross-sectional studies that used linear mixed models to explore the association between the compliance of 24-hour movement behavior recommendations and cognitive ability, including emotion comprehension, theory of mind [22] and EF [23]. Nevertheless, this method has some shortcomings. Firstly, this method can only roughly divide the population into those who meet or do not meet the recommended guidelines, which limits the specific

guidance for intervention measures. Secondly, this method ignores the feature of 24-hour movement behaviors as compositional data. Specifically, compositional data convey the relative reliability of time rather than the absolute quantity. The use of traditional linear regression analysis may lead to issues such as pseudo-correlation and multi-collinearity between compositional data [24].

Recently, researchers have proposed and adopted a new analytical method, namely the compositional data isotemporal substitution model [25], to examine the relationships between 24-hour movement behaviors and health indicators and the impact of time redistribution of various movement behaviors on health indicators. A study from low-income areas in Brazil revealed a correlation between 24-hour movement behaviors and EF in preschool children. Among them, replacing sleep and LPA with MVPA is associated with a significant improvement in EF [26]. On this basis, another study tested the relationships between 24-hour movement behaviors and the three subcomponents of EF (inhibitory control, working memory, and cognitive flexibility) and reported that 24-hour movement behaviors are significantly correlated with inhibitory control and working memory. Among them, replacing sleep and SB with MVPA is related to improving inhibitory control, whereas replacing sleep, SB, and MVPA with LPA is related to improving working memory [27]. However, there is currently no research analyzing the relationship between 24-hour movement behaviors and cognitive ability such as the intelligence quotient (IQ) in preschool children [28].

In summary, this study adopted the compositional data isotemporal substitution method to further explore the comprehensive impact of 24-hour movement behaviors on the cognitive ability of preschool children. This approach enabled us to examine the dynamic compositions of movement behaviors within a daily cycle and their influences on the cognitive ability of preschool children. The present study, therefore, offers a more holistic perspective regarding the optimal composition of 24-hour movement behaviors and insights into strategies for facilitating cognitive development in preschool children via intervention in their 24-hour movement behaviors. We aimed to examine 1) the relationships between the 24-hour movement behaviors and cognitive ability; 2) the expected changes in the cognitive ability of preschool children with the time reallocation of 24-hour movement behaviors (for example, reallocating 15 min of LPA to MVPA); and 3) the dose-response relationships between the time allocation of various movement behaviors and the cognitive ability of preschool children.

## 1. Method

### 1.1 Design and participants

This was a cross-sectional study of preschool children aged 3-6 years, regardless of sex. This study invited the Zhuzhou Early Childhood Education Association to register kindergartens for evaluation. The Preschool education zone in Zhuzhou City Urban Area is organized into 4 sectors. In order to ensure a representative sample across the city's diverse educational sectors, 4 kindergartens were strategically selected for this study. After screening through questionnaires and health assessments, participants were eligible to participate in the study. The participants were recruited on the basis of the following inclusion criteria: 1) had not been diagnosed with any related diseases that hinder regular PA and 2) were in normal development and possessed corresponding cognitive ability.

The number of participants required in the study was estimated using G*Power (3.1.9). This study used a multiple linear regression model to explore the association between sex, age, four types of movement behaviors, and cognitive abilities in preschool children. Therefore, to achieve a medium effect size (Cohen $f^2$) of 0.15, with an alpha of 0.05, 95% statistical power in the multiple linear regression model [29], 129 participants were required to ensure a robust statistical analysis.

A total of 260 registered preschool children were invited to participate in this study. Prior to the study, researchers provided a written consent form to all parents of preschool children. During the research process, participants' parents or legal guardians may withdraw their consent to participate in this study at any time without explanation. Forty-two households did not provide written consent. During the evaluation process, PA, sleep time and SB data were missing for 27 children. In the final sample, there were 78 boys and 113 girls (M = 4.95 years old; SD = 0.98) (Fig 1).

## 1.2 Measures

**1.2.1 Movement behaviors.** PA and SB were objectively measured using a tri-axial accelerometry (Actigraph, model GT3-BT). Actigraph GT3 accelerometers had been widely used in previous studies with preschoolers [30]. Before the measurement, kindergarten teachers and parents received written and video instructions on the correct use of accelerometers. In addition, parents were also required to record the daily wearing and removal times of their children's accelerometers in an activity diary. The teacher checked the children's wearing of physical activity accelerometers during their time in the kindergarten every day. The device initialization settings and information entry were performed using the ActiLife software (version 6.13.4).

The accelerometer was fixed on the right hip for 24 hours for seven consecutive days, but the instrument had been removed for water activities, bathing, and sleeping at night. Participants should not change their daily activity habits while wearing them, and should maintain their daily activities. The sampling interval of the accelerometer is set to 15 s, with an effective wearing time of at least three days (two weekdays and one weekend day) and at least 8 hours of wearing per day [31]. The non-wearing time was defined using the Choi algorithm[32], with zero counts for 90 consecutive minutes. During this period, a maximum of 2 minutes from 0 to 100 counts/min was allowed. The accelerometer was initialized at a sampling rate of 30 Hz and then reintegrated into 15-s epochs for analysis. Butte's et al.[33] cut-off point were used to classify the activity count of the accelerometer into different intensities. These cut-off points were SB: < 239 counts per minute (CPM); light: 240 ~ 2119 CPM; moderate: 2119 ~ 4449 CPM and vigorous: ≥ 4450 CPM. The duration of MVPA was calculated by adding moderate physical activity (MPA) and vigorous physical activity (VPA) times. The accelerometer

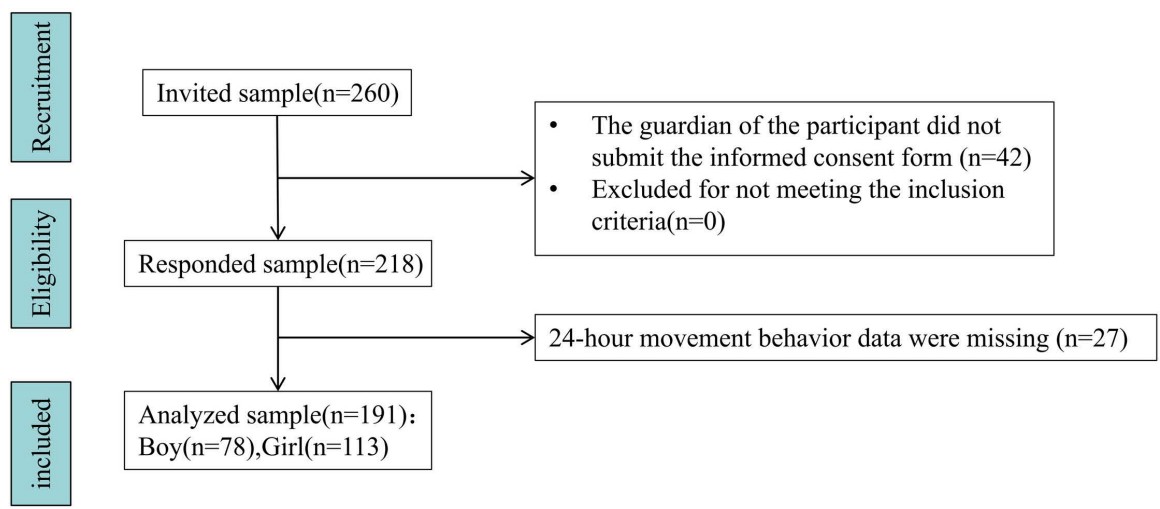

**Fig 1. Recruitment process diagram for subjects.**

data recording was set to start at 0:00 on the second day after issuance. After the instrument was collected, ActiLife (Version 6.13.4) software was used to export and analyze the accelerometer data and initialize the accelerometer.

The sleep time of preschool children in this study is the sum of their daytime and nighttime bedtime to wake-up time. The bedtime and wake-up times during the weekday are recorded by the childcare teacher by taking photos and sending them to the WeChat group. The children's nighttime sleep time and weekend daytime sleep time were reported by the parents, who record the children's nighttime sleep time and weekend daytime sleep time for a week in the sleep log. The sleep log was used to record the time when children wake up and fall asleep every day, seven days a week. Research had shown a good correlation between sleep time recorded in sleep logs and recorded in accelerometers [34, 35]. The weighted average of sleep time on weekdays and weekends was calculated as follows: [(sleep time on weekdays) × effective days + (sleep time on weekend) × effective days]/ total effective days.

**1.2.2 Cognitive ability.** Considering the large number of participants, young age, and numerous the C-WYCSI test items in this study, the completion time of the test was relatively long. Preschool children did not be able to concentrate for a long time to complete the test, leading to errors in the research results. Therefore, the cognitive ability of the participants in this study were evaluated by two trained researchers using a short form of C-WYCSI, and this simplified test was also widely adopted in previous studies investigating cognitive function [36–38]. The simplified test consists of four subtests: the Information subtest, the Vocabulary subtest, the Picture Completion subtest, and the Block Design subtest. Among them, Information subtest and Vocabulary subtest are examples of Verbal Intelligence Quotient (VIQ) tests, while Picture Completion subtest and Block Design subtest belong to the Performance Intelligence Quotient (PIQ) tests. The Information subtest required participants to answer questions related to daily knowledge. Answering correctly earned 1 point, whereas answering incorrectly earned 0 points (total score ranged from 0–23). The Vocabulary subtest required young children to recognize the proper answer from four pictures corresponding to the words indicated by the tester. A score of 1 point was given for correct answers, and 0 points were given to an error answer, with a total score ranging from 0–44 points. The Picture Completion subtest required young children to recognize and point out the missing parts in the picture. Answering correctly earned 1 point, answering incorrectly earned 0 points, and the total score ranged from 0–25 points. The Block Design subtest required children to use existing wooden blocks to recreate the patterns on the booklet as quickly as possible. Testing was stopped after two consecutive 0 points. The scores for patterns 1–6 were divided into three levels: 2 points, 1 point, and 0 points. The first attempt of each pattern was correctly placed within the time limit, with a score of 2 points; The second attempt was to correctly pose within the time limit and score 1 point; Failed twice and received 0 point. The 7th to 9th patterns, completed within 15 seconds for the first attempt, earned 1 point; The 10th and 11th patterns, completed within 15 seconds of the first attempt, earned 2 points, and those completed within 45 seconds earned 1 point. The total score was 0–29 points.

Upon completion of the test, the original scores were converted into scale scores according to C-WYCSI's operating manual. The scores of the VIQ and PIQ scales were equal to the sum of the scores of the two verbal subtest scales and the sum of the scores of the two performance test scales, respectively. The Full Intelligence Quotient (FIQ) was evaluated on the basis of the weighted scores of each subtest in the operating manual of C-WYCSI. The operation manual divides the total IQ scores of preschool children into five categories: significantly lower than normal (<70 points), slightly lower than normal (70–90 points), normal (90–110 points), somewhat higher than normal (110–130 points), and significantly higher than normal (>130 points). The test took about 30 minutes to complete and has been widely adopted in previous studies [14,39].

## 1.3 Procedures

With ethics approval from the lead author's institution, a 2-hour seminar was held in each invited kindergarten, during which all kindergarten staff, researchers, participating children and their parents or legal guardians were informed of the purpose, protocol, and procedures of the study and agreed to participate. In addition, a 3-hour training course was held for all staff and researchers, providing detailed explanations and practical experience with the use of instruments and measurements.

To avoid collecting data in months where participants may have unusual PA patterns, such as summer and winter, this study evaluated and selected March to May 2023. All the children were authorized by their parents through consent forms. The kindergarten provided all sociodemographic data (child gender, age, date of birth, parental contact information, and address). The parents were invited to the school for a meeting and were interviewed individually. In this interview, sleep logs were distributed to parents to record their children's sleep time at night and wakefulness time in the morning for seven consecutive days. Owing to the long-term nature of measuring movement behaviors, researchers, teachers, and parents had jointly established WeChat groups to facilitate emergency contact with parents. On the day the accelerometers were distributed, the cognitive ability of the preschool children was assessed, and the accelerometers were placed on the participating children who used them continuously for eight days (the accelerometers were recorded from the second day onward). The participants were asked to continue their normal daily lives every day. The rewards for participants include printed reports of IQ scores, physical fitness, anthropometric assessments, and other results. In addition, children received a small toy, and their parents received some essential items such as toilet paper or laundry detergent.

## 1.4 Ethics statement

The studies involving humans were approved by Hunan University of Technology Institutional Review Board (no. 20230007).

## 1.5 Statistical analyses

Compositional data analyses (CoDA) were performed using R version 3.6.3 software (R Development Core Team). Following recommendations [25], standard and compositional descriptive statistics were calculated for comparison. An isometric log-ratio (ILR) transformation was applied [40]. This approach transferred compositional data from the constrained simplex space to the unconstrained real space and facilitated standard statistical analysis. The variance matrix was used to describe the dispersion of daily composition [25]. Because the fact that the variance of individual composition cannot reflect the interdependence of movement behaviors, it was necessary to calculate pairwise logarithmic variance to describe the dispersion trend between compositional data. The smaller the coefficient of variance was, the more consistent the ratio between the two movement behaviors [25].

A multiple linear regression model was used to test the relationship between 24-hour movement behavior (independent variable) and cognitive ability (dependent variable). The covariates (age and sex) were used as independent variables [41]. The dependent variables were VIQ, PIQ, and FIQ, respectively. The specific multiple linear regression models are as follows [40]:

$$E(Y|Z) = \beta_0 + \beta_1 Z_1 + \beta_2 Z_2 + \beta_3 Z_3 + \cdots + \beta_{d-1} Z_{d-1} + sex + age \tag{1}$$

$$Z_i = \frac{\sqrt{d-i}}{\sqrt{d-i+1}} \ln\left(\frac{b_i}{\sqrt[d-i]{\prod_{j=i+1}^{d} b_j}}\right) \tag{2}$$

The isotemporal substitution model was used to predict the differences in outcome indicators related to the replacement of two movement behaviors with the same duration while the duration of other movement behaviors remained constant [42]. Specifically, we used the geometric mean of the sample compositions as baseline data and subsequently created a new component composition, followed by subtraction calculations to identify the estimated differences between the baseline composition and the new composition in predicting cognitive outcomes. On the basis of recommendations [21] that a 15-minute change in movement behavior significantly affected health outcome indicators, a 15-minute reassignment between two movement behaviors was adopted to reflect the estimated difference in cognitive ability outcomes. The dose-response effect of the mutual reallocation of movement behaviors and cognitive ability was examined in increments of 5 min and continued for a longer period of up to 30 min.

## 2. Result

### 2.1 Descriptive statistics.

The descriptive statistics of the participants are shown in Table 1. The 24-hour movement behavior compositional data of preschool children are shown in Table 2. On average, participants underwent 608.7 min (42.3%) of sleep, 437.6 min (30.4%) of SB, 358.1 min (24.9%) of LPA, and 35.6 min (2.4%) of MVPA per day. All pairwise log-ratio variances are summarized in the variation matrix (Table 3). The variance of the log in SB and sleep was closest to 0 (in SB/sleep = 0.046), indicating that these two activities were the most highly co-dependent. The

**Table 1. Descriptive statistical analysis of related variables.**

| Variable | M(SD)/N(%) |
|---|---|
| Age | 4.59 ± 0.99 |
| Sex | |
| Male | 78(40.8%) |
| Female | 113(59.2%) |
| Cognitive ability | |
| VIQ | 73.9 ± 8.1 |
| PIQ | 68.5 ± 6.5 |
| FIQ | 105.9 ± 12.1 |

**Table 2. Descriptive statistical analysis of related variables.**

| Movement behavior | Sleep (min/d) | SB (min/d) | LPA (min/d) | MVPA (min/d) |
|---|---|---|---|---|
| Arithmetic mean | 608.7 | 437.6 | 358.1 | 35.6 |
| Arithmetic mean percentage (%) | 42.3 | 30.4 | 24.9 | 2.4 |
| Composition mean percentage (%) | 42.6 | 30.4 | 24.8 | 2.2 |

**Table 3. Variation matrix of the compositional data.**

| | Sleep | SB | LPA | MVPA |
|---|---|---|---|---|
| Sleep | 0 | 0.035 | 0.047 | 0.343 |
| SB | 0.035 | 0 | 0.077 | 0.384 |
| LPA | 0.047 | 0.077 | 0 | 0.247 |
| MVPA | 0.343 | 0.384 | 0.247 | 0 |

variance of the log in MVPA and SB was the largest (in MVPA/SB = 0.143), indicating that low co-dependent between MVPA and SB.

## 2.2 Association between 24-hour movement behavior compositions and cognitive ability

The linear regression analysis between 24-hour movement behavior and cognitive ability is shown in Table 4. The results indicate that 24-hour movement behavior is significantly correlated with VIQ ($R^2 = 0.19$, P < 0.001), PIQ ($R^2 = 0.15$, P < 0.001), and FIQ ($R^2 = 0.13$, P < 0.001). Specifically, MVPA was significantly positively correlated with PIQ ($\beta(4) 1 = 5.09$, P < 0.05) and FIQ ($\beta(4) 1 = 8.85$, P < 0.05). LPA was significantly positively correlated with PIQ ($\beta(3) 1 = 5.66$, P < 0.05) and FIQ ($\beta(3) 1 = 9.64$, P < 0.05). Sleep was significantly negatively correlated with VIQ ($\beta(1) 1 = -10.08$, P < 0.05), PIQ ($\beta(1) 1 = -9.89$, P < 0.05), and FIQ ($\beta(1) 1 = -19.77$, P < 0.05). However, there was no significant correlation between SB and cognitive ability.

## 2.3 Predictions for reallocation of time

Compositional isotemporal substitution was carried out for cognitive ability outcome measures. The results of the reallocation of 15 min between movement behaviors are displayed in Table 5. Reallocating 15 min from SB to MVPA was associated with predicted increases in VIQ (1.04 units), PIQ (0.91 units), and FIQ (1.98 units), whereas reallocating 15 min from MVPA to SB was associated with a predicted decrease in VIQ (1.55 units), PIQ (1.40 units), and FIQ

**Table 4. Linear regression analysis with compositional between 24-h movement behavior and cognitive ability.**

|  | Sleep | | | SB | | | LPA | | | MVPA | | | Model |
|---|---|---|---|---|---|---|---|---|---|---|---|---|---|
|  | β(1) 1 | 95%Cl | P | β(2) 1 | 95%Cl | P | β(3) 1 | 95%Cl | P | β(4) 1 | 95%Cl | P | P(R2) |
| VIQ | **−10.08** | [−18.79, −1.46] | 0.02 | 2.10 | [−4.28, 8.49] | 0.52 | 4.00 | [−1.50, 9.52] | 0.15 | 3.97 | [−1.37, 9.32] | 0.14 | <0.001(0.19) |
| PIQ | **−9.89** | [−16.89, −2.89] | 0.006 | −0.86 | [−6.05, 4.23] | 0.74 | **5.66** | [1.19, 10.14] | 0.01 | **5.09** | [0.75, 9.43] | 0.02 | <0.001(0.15) |
| FIQ | **−19.77** | [−32.98, −6.57] | 0.004 | 1.29 | [−8.49, 11.07] | 0.80 | **9.64** | [1.20, 18.08] | 0.03 | **8.85** | [0.65, 17.03] | 0.03 | <0.001(0.13) |

Statistically significant associations at the 95% confidence interval (CI) level (p < 0.05) are highlighted in bold.

**Table 5. Estimated difference in the total scores of cognitive ability for 15 min isotemporal substitution between 24-hour movement behaviors.**

| Add | Remove | VIQ (95% CI) | PIQ (95% CI) | FIQ (95% CI) |
|---|---|---|---|---|
| Sleep | SB | 0.49(0.14,0.84)* | 0.39(0.10,0.68)* | 0.87(0.33,1.41)* |
| Sleep | LPA | 0.41(0.02,0.79)* | 0.48(0.17,0.80)* | 0.88(0.29,1.47)* |
| Sleep | MVPA | −1.07(−2.60,0.46) | −1.01(−2.26,0.24) | −2.13(−4.49,0.23) |
| SB | Sleep | −0.49(−0.83,−0.14)* | −0.39(−0.67,−0.11)* | −0.87(−1.41,−0.33)* |
| SB | LPA | −0.08(−0.42,0.27) | 0.10(−0.19,0.38) | −0.02(−0.52,0.56) |
| SB | MVPA | −1.55(−3.06,−0.04)* | −1.40(−2.63,−0.16)* | −2.99(−5.32,−0.66)* |
| LPA | Sleep | −0.40(−0.82,−0.03)* | −0.48(−0.79,−0.17)* | −0.87(−1.45,−0.29)* |
| LPA | SB | 0.09(−0.25,0.43) | −0.08(−0.36,0.20) | 0.02(−0.52,0.54) |
| LPA | MVPA | −1.46(−3.16,0.23) | −1.48(−2.87,−0.10)* | −2.99(−5.60,−0.37)* |
| MVPA | Sleep | 0.54(−0.38,1.47) | 0.52(−0.24,1.27) | 1.09(−0.34,2.53) |
| MVPA | SB | 1.04(0.13,1.95)* | 0.91(0.17,1.66)* | 1.98(0.57,3.38)* |
| MVPA | LPA | 0.96(−0.14,2.05) | 1.00(0.11,1.90)* | 1.98(0.29,3.68)* |

all models are adjusted for age and sex.

* P < 0.05.

(2.99 units). Reallocating 15 min from LPA to MVPA was associated with predicted increases in the PIQ (1.00 units) and FIQ (1.98 units), whereas reallocating 15 min from MVPA to LPA was associated with a predicted decrease in PIQ (1.48 units) and FIQ (2.99 units). whereas reallocating 15 min from sleep to LPA was associated with predicted decreases in VIQ (0.40 units), PIQ (0.48 units), and FIQ (0.87 units), while Reallocating 15min from LPA to sleep was associated with a predicted decrease in VIQ (0.41 units), PIQ (0.48 units), and FIQ (0.88 units). Reallocating 15 min from sleep to SB was associated with predicted increases in VIQ (0.49 units), PIQ (0.39 units), and FIQ (0.87 units), whereas reallocating 15 min from sleep to SB was associated with a predicted decrease in VIQ (0.49 units), PIQ (0.39 units), and FIQ (0.87 units). However, these findings are preliminary and highlight the need for further research to understand the causal relationships.

We further examined the dose-response relationships between the redistribution of 24-hour movement behaviors and VIQ (Fig 2), PIQ (Fig 3), and FIQ (Fig 4), with increments of 5 min and durations extended to 30 min. The reallocations from SB, sleep, and LPA to

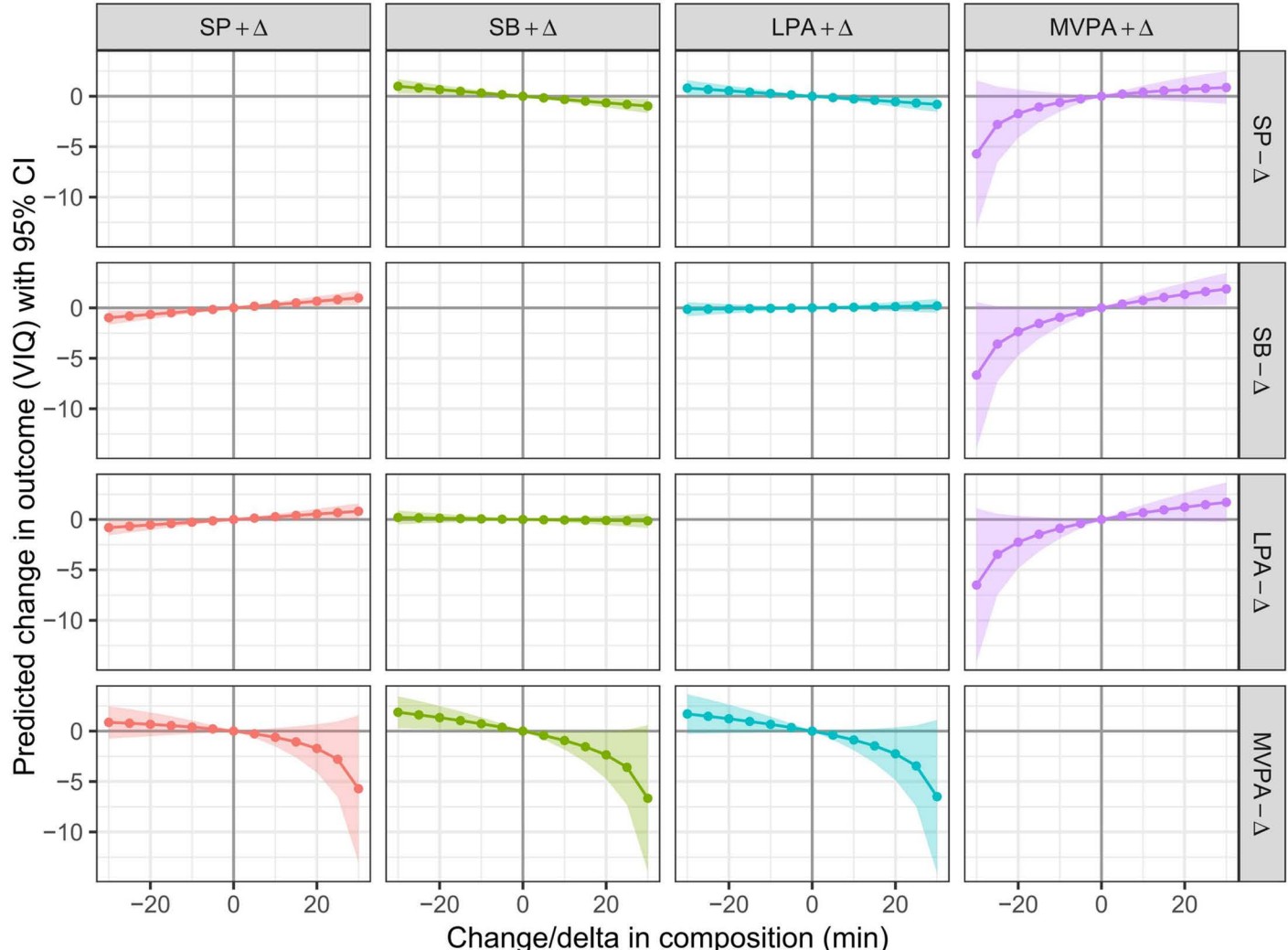

**Fig 2. Estimated changes (95% CI) in VIQ for −30 to 30 min isotemporal substitution between MVPA and other movement behaviors.** All the data were adjusted for age and sex.

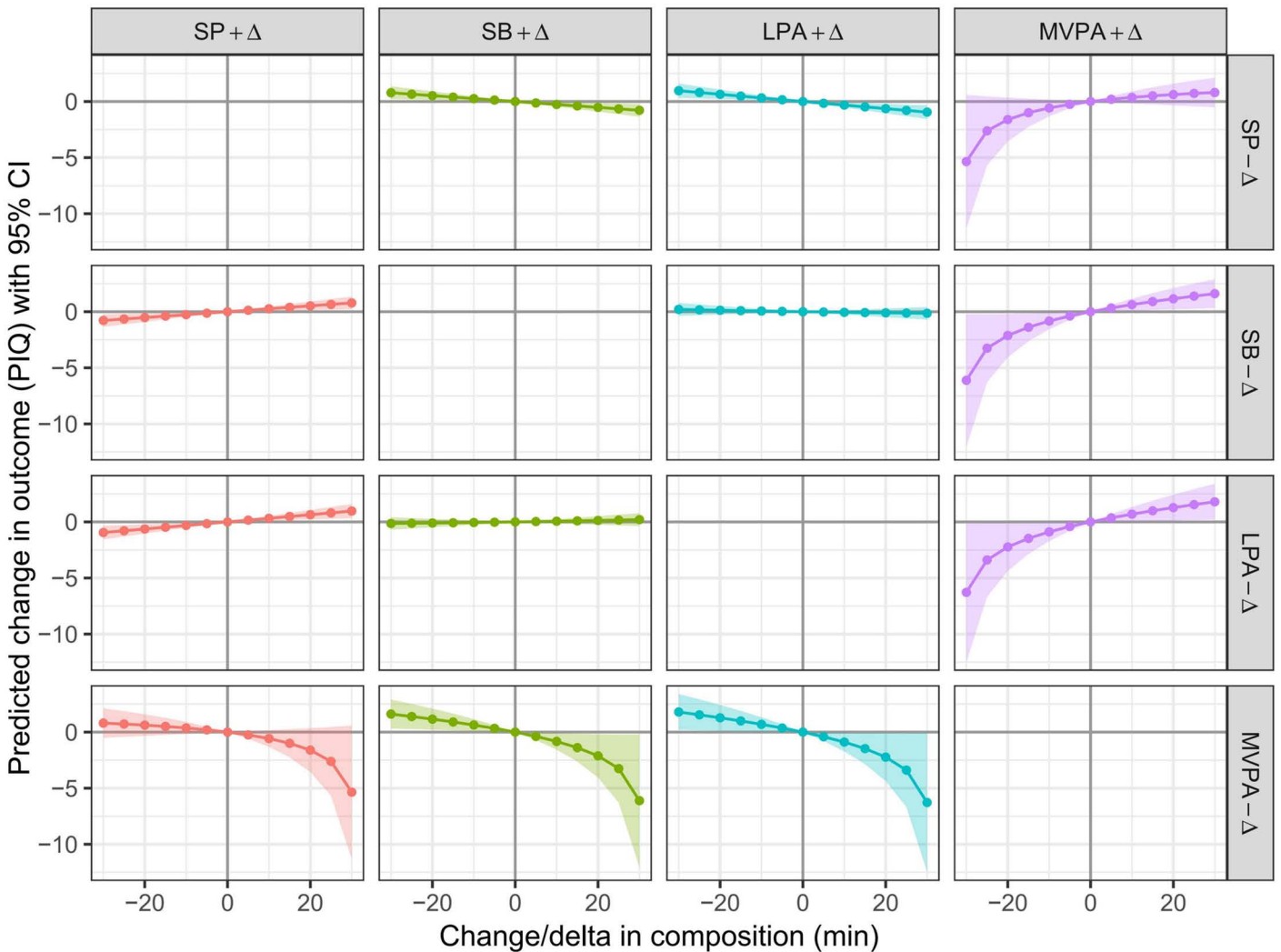

**Fig 3. Estimated changes (95% CI) in PIQ for −30 to 30 min isotemporal substitution between MVPA and other movement behaviors.** All the data were adjusted for age and sex.

MVPA were associated with better VIQ, PIQ, and FIQ, whereas the reallocations from MVPA to SB, sleep, and LPA were associated with poorer VIQ, PIQ, and FIQ. This association was asymmetrical. The negative impact of reallocating time from MVPA to SB, sleep, and LPA outweighs the benefits of increasing MVPA time at the expense of SB, sleep, and LPA.

## 3. Discussion

The purpose of this study was to utilize compositional data analysis to explore the relationships between the 24-hour movement behavior patterns and the cognitive ability of Chinese preschool children. In addition, this study aims to examine the effects of isotemporal substitution of 24-hour movement behavior on cognitive ability outcomes. This study revealed that preschool children spent 42.3% of the 24-hour daily cycle in sleep, 30.4% in SB, 24.9% in LPA, and 2.4% in MVPA. Compared with previous studies, the MVPA and sleep of preschool children in this study were shorter, and the LPA was longer [27,31]. The 24-hour movement behavior of preschool children is influenced by various factors, such as the family

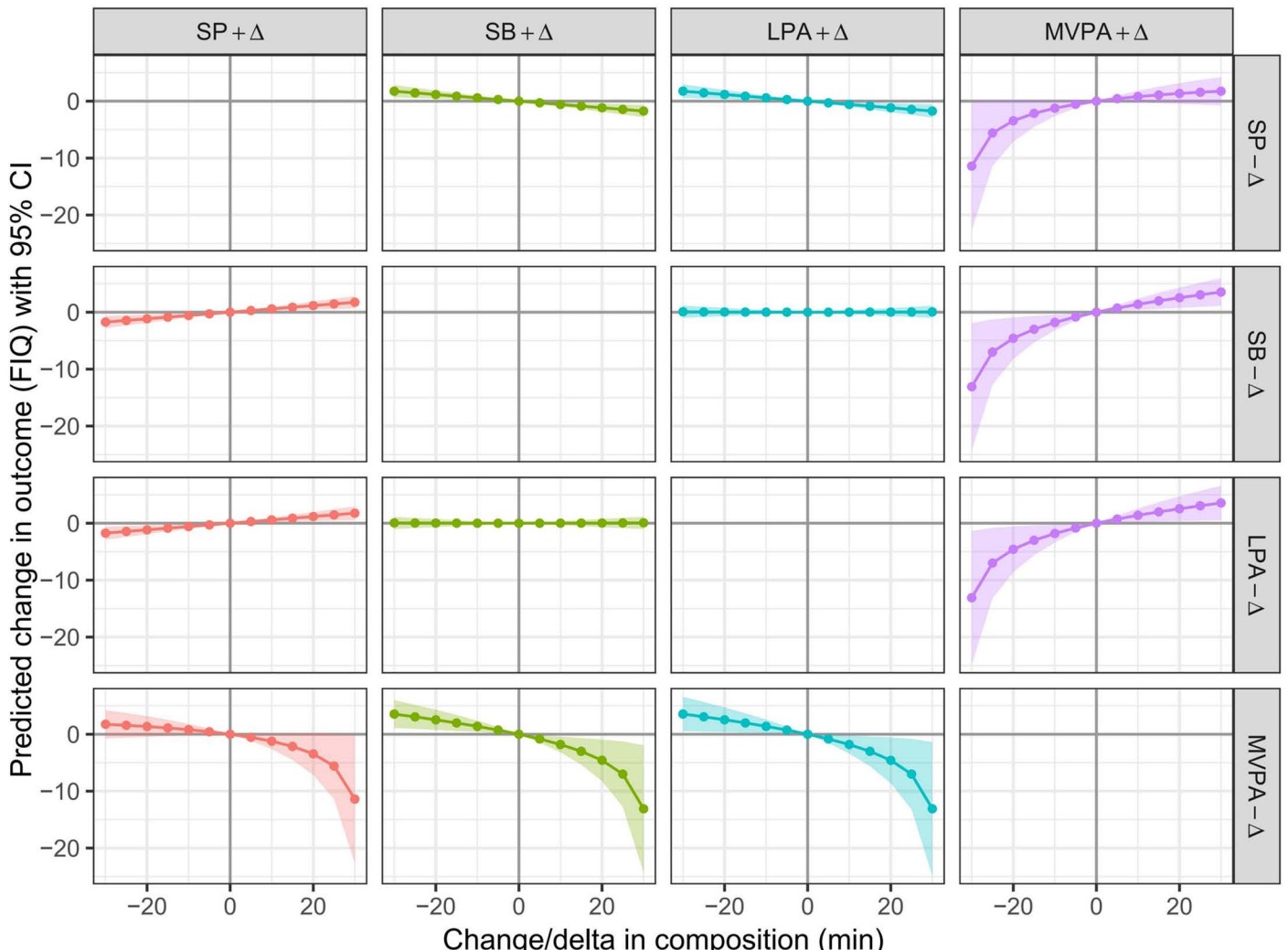

**Fig 4. Estimated changes (95% CI) in FIQ for −30 to 30 min isotemporal substitution between MVPA and other movement behaviors.** All the data were adjusted for age and sex.

environment [43] and climate factors [44]. A review report suggests that the sleep model of preschool children develops in the family environment and that the quality of parent-child interaction and parental emotions could impact the sleep of preschool children. The healthy behavior of the mother and the available space in the room are significant correlated with PA in preschool children [45]. In addition, review reports have suggested that extreme weather can have sustained adverse effects on PA [44]. Notably, during the 24-hour movement behavior period of the preschool children surveyed in this study, there were more severe weather conditions and preschool children often engaged in PA indoors. This may be the reason for the insufficient MVPA in preschool children in this study.

The main findings of this study indicate that when considering all individual behaviors as a whole of 24-hour movement behavior could significantly explain 19%, 15%, and 13% of the changes in VIQ, PIQ, and FIQ, respectively. The reallocation of the 24-hour movement behavior time is related to changes in VIQ, PIQ, and FIQ. Specifically, dose-response analysis revealed that the effects of isotemporal substitution between MVPA, SB, sleep, and LPA on

VIQ, PIQ, and FIQ were asymmetric. The evidence on the relationship between PA and EF in preschool children in traditional research has been divergent in the past. Willoughby et al. [46] reported a negative correlation between MVPA and EF in preschool children, whereas LPA was not significantly correlated with EF. A study using traditional multiple linear regression analysis on the relationship between PA and EF in preschool children revealed a negative correlation between LPA and inhibitory control [47]. Quan et al. [14] reported that the VIQ, PIQ, and FIQ of boys were significantly correlated with LPA rather than MVPA, whereas the relationship between PA and female cognitive ability was not significant. In contrary, our study revealed a significant positive correlation between MVPA and LPA in preschool children and their PIQ and VIQ scores. Our study is consistent with the research findings of Bezerra et al. [26], who reported both MVPA and LPA positively affect the cognitive ability of preschool children through the comprehensive effect of 24-hour movement behavior. This finding reinforces the research evidence that exercise promotes cognition. However, the results of the isotemporal substitution model revealed that reallocating LPA to MVPA had a significant positive correlation with the PIQ and FIQ in preschool children. Previous studies using compositional data analysis have also reported similar findings that reallocating LPA to MVPA results in significant positive correlation between inhibition and working memory in preschool children [26, 27].This suggests that PA intensity may be an important factor affecting the cognitive development of preschool children. When considering the PA intensity of preschool children, the stronger the performance of MVPA in different areas of cognitive ability compared to LPA. On the other hand, previous studies using traditional linear regression models have limitations. The sum of all movement behavior times in a day is 24 hours, and a change in one movement behavior time will inevitably affect one or more other activity times. Therefore, it is necessary to view 24-hour movement behavior as a whole, and in the future, relevant research needs to be conducted using compositional data analysis methods.

Our study revealed that, compared with other movement behaviors, sleep significantly negatively affects the VIQ, PIQ, and FIQ of preschool children. However, previous evidence has shown that sufficient sleep has a positive predictive effect on cognitive ability [7]. In addition, Preschool children with insufficient sleep have poorer cognitive ability, whereas those with appropriate total sleep have better cognitive development [7,48]. Li et al. [31] reported a significant negative correlation between sleep and scores of prosocial behavior scores in preschool children. Although the results of our study were the opposite of those of the aforementioned studies, this does not mean that sleep has a negative effect on the cognitive ability of preschool children. This may be because the sleep compliance rate of preschool children surveyed in this study (according to the WHO's 24-hour movement behavior guidelines for 0–5-year-old) [49] was only 59.1%. Compared with the other three movement behaviors, sleep negatively affects preschool children's cognitive ability. The results of isotemporal substitution (Table 4) indicate that reallocating SB and LPA to sleep has a significant positive effect on the cognitive ability of preschool children, surpassing the adverse impacts of increasing SB and LPA time at the cost of sleep. This result emphasizes the importance of sleep in the cognitive development process of preschool children. Although numerous studies have confirmed the promoting effects of sleep, non-screen-based SB (NSCSB), and LPA on cognition, considering the interactions between different types of motor behaviors and their cumulative effects on cognitive ability may be more complex than previously understood [7,14].

The correlation between SB and the cognitive ability of preschool children was not significant compared with that between SB and the other three movement behaviors in our study. The reason for this may be related to the type of SB. A review suggested that the effects of different types and task requirements of SB on the cognitive ability of preschool children are not consistent [17]. For example, screen-based SB (SCSB) is not conducive to the development

of cognitive ability in preschool children [9,50], whereas NSCSB is beneficial for the development of cognitive abilities in preschool children, such as reading and drawing, in preschool children [51]. The accelerometer used in this study can only record the SB time and cannot distinguish the type of SB.

The dose-response (Figs 1–3) results revealed that there was asymmetry in the changes of FIQ in preschool children when MVPA and other movement behaviors were replaced with each other. This feature has also been explored in other studies on the impact of 24-hour movement behavior on the health indicators of preschool children [27]. This characteristic is related to the proportion of MVPA's 24-hour movement behavior time. When MVPA is assigned to other movement behaviors in a "one-by-one" manner on the basis of an absolute duration of 15 min, the substitution effect is significant, because the 15 min MVPA accounts for 42.1% of the total MVPA time (35.6 min), whereas sleep, SB, and an increase in LPA by 15 min account for only 2.5%, 3.4%, and 4.2% of the total time, respectively, resulting in a suppressed substitution effect. Therefore, when the other three movement behaviors are replaced with a 15-minute MVPA isotemporal duration, the change in the duration of the other three movement behaviors is relatively small, and the effect on the cognitive development of preschool children is also relatively small. When other movement behaviors replace MVPA within 15 min, there is a significant change in MVPA time, which leads to a substantial change in the VIQ, PIQ and FIQ of preschool children. This feature further illustrates the importance of MVPA in the cognitive development of preschool children. Therefore, it is necessary to pay close attention to the negative impact of insufficient MVPA on preschool children's cognitive development.

Our research has several advantages. Firstly, using objective measurements based on PA and SB avoids recall bias from parents or teachers toward proxy reports. Secondly, although time data of sleep are still based on subjective measurements, sleep logs require parents to record their children's SP time in a timely manner on a daily basis, which effectively reduces long-term recall bias compared with other subjective measurement forms using questionnaires. Finally, the use of compositional data analysis and isotemporal substitution models to evaluate the relationship between 24-hour movement behavior and preschool children's cognitive ability strengthened this study's internal validity.

## 4. limitations

However, this study has certain limitations that must be considered. Firstly, these findings are based on cross-sectional data and do not allow for tracking the persistence and consistency of movement behavior over time. In the future, longitudinal or randomized controlled studies are needed. Secondly, the evaluation of SB is not comprehensive and does not consider the types of activities that children engage in during the SB time. Future research should consider SB patterns and activities. Finally, this analysis controlled for only age and sex, without controlling for other factors that may affect cognitive ability. Parent-child interaction plays a crucial role in the acquisition of cognitive ability in preschool children [45]. The socioeconomic status of preschool children is related to their cognitive ability [52]. In addition, the family environment is also a critical factor affecting the allocation of 24-hour movement behavior time [43], and future research should consider these factors when studying the relationship between 24-hour movement behavior and cognitive ability.

## 5. Conclusion

The use of combinatorial paradigms can help us better understand the connection between 24-hour movement behavior and cognitive ability in preschool children. Current research

shows that when we study movement behavior as a continuous spectrum, the amount of time a child spends on different activities can accurately predict their cognitive ability. Furthermore, increasing the amount of time spent on MVPA or sleep while decreasing the amount of time spent on SB and LPA can lead to positive changes in cognitive ability. Specifically, increasing MVPA while decreasing LPA is associated with improved performance in PIQ and FIQ. This highlights the importance of ensuring that preschool children engage in enough MVPA and sleep for optimal cognitive development.

## Supporting information

**S1 Raw data.  Basic information, 24-hour movement behavior time data, and cognitive ability score of preschool children.**
(XLS)

## Acknowledgements

We would like to thank the participants in this study.

## Author contributions

**Conceptualization:** Shiqiang Wang.

**Data curation:** Zitong Ma.

**Formal analysis:** Zhihan Xu.

**Funding acquisition:** Shiqiang Wang.

**Investigation:** Zhihan Xu, Zitong Ma, Dan Li.

**Methodology:** Zhihan Xu.

**Project administration:** Dan Li.

**Supervision:** Shiqiang Wang.

**Validation:** Shiqiang Wang.

**Writing – original draft:** Zhihan Xu.

**Writing – review & editing:** Zhihan Xu, Shiqiang Wang, Shuge Zhang.

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
