## [Decision Letter · Decision Letter 0]

3 Dec 2024

PONE-D-24-2892724-hour movement behaviors and cognitive ability in pre-school children: A compositional and isotemporal reallocation analysisPLOS ONE

Dear Dr. Wang,

Thank you for submitting your manuscript to PLOS ONE. After careful consideration, we feel that it has merit but does not fully meet PLOS ONE’s publication criteria as it currently stands. Therefore, we invite you to submit a revised version of the manuscript that addresses the points raised during the review process.

We look forward to receiving your revised manuscript.

Kind regards,

Jindong Chang, Ph.D.

Academic Editor

PLOS ONE

Journal Requirements:

4. Thank you for stating the following financial disclosure: “Hunan Provincial Education Science Planning Project Base Project, grant number XJK23AJD055”.

5. We note that you have indicated that there are restrictions to data sharing for this study. PLOS only allows data to be available upon request if there are legal or ethical restrictions on sharing data publicly. For more information on unacceptable data access restrictions, please see http://journals.plos.org/plosone/s/data-availability#loc-unacceptable-data-access-restrictions. Before we proceed with your manuscript, please address the following prompts: a) If there are ethical or legal restrictions on sharing a de-identified data set, please explain them in detail (e.g., data contain potentially identifying or sensitive patient information, data are owned by a third-party organization, etc.) and who has imposed them (e.g., a Research Ethics Committee or Institutional Review Board, etc.). Please also provide contact information for a data access committee, ethics committee, or other institutional body to which data requests may be sent. b) If there are no restrictions, please upload the minimal anonymized data set necessary to replicate your study findings to a stable, public repository and provide us with the relevant URLs, DOIs, or accession numbers. For a list of recommended repositories, please see https://journals.plos.org/plosone/s/recommended-repositories. You also have the option of uploading the data as Supporting Information files, but we would recommend depositing data directly to a data repository if possible. We will update your Data Availability statement on your behalf to reflect the information you provide.

Reviewers' comments:

Reviewer's Responses to Questions

**Comments to the Author**

1. Is the manuscript technically sound, and do the data support the conclusions?

Reviewer #1: Yes

Reviewer #2: Partly

Reviewer #3: Yes

2. Has the statistical analysis been performed appropriately and rigorously? 

Reviewer #1: Yes

Reviewer #2: Yes

Reviewer #3: Yes

3. Have the authors made all data underlying the findings in their manuscript fully available?

Reviewer #1: No

Reviewer #2: Yes

Reviewer #3: Yes

4. Is the manuscript presented in an intelligible fashion and written in standard English?

Reviewer #1: Yes

Reviewer #2: Yes

Reviewer #3: Yes

5. Review Comments to the Author

Reviewer #1: This is an interesting study conducted in a relatively under-studied population (pre-schoolers 24-h movement behaviours). Generally it is well written and argued. Please find some suggestions below:

Introduction:

Please rephrase “This is due to the subpar quality of 49 the studies and the limited accumulation of research evidence.” Or provide evidence for why the 49 studies are “subpar” – what does this mean?

For the statement “Moreover, different sleep parameters also have differences in the development of cognitive abilities in early children [11]” please provide some context of what these sleep parameters are. Will they be explored in this study?

“Similarly, the association between SB and cognitive development in early children should also vary depending on the type of sedentary behavior [12].” Can you please provide some examples here.

“The advantage of this analysis method is that it fully corresponds to the requirements of the behavioral guidelines, with satisfaction marked as 1 and dissatisfaction marked as 0, resulting in a clear and intuitive presentation of the data.” This is confusing to me. Does satisfaction mean “compliance with the guidelines” and how does this work when there are multiple guidelines each for different behaviours within the 24 h day, i.e., it’s possible to meet some guidelines (sleep and PA), and not meet others (screen time).

“A study from low-income areas in Brazil found a positive correlation between 24-hour movement behaviors and EF in pre-school children” This doesn’t make sense – it’s not possible to have a positive correlation between all 24-h MBs and an outcome? Do you mean there was an association between the composition and EF?

Please refine the Aims

Aim 1: i think this is to evaluate the associations between the overall 24-h MB composition and outcomes?

Aim 2: please reword to avoid inference of directionality which is not possible with cross-sectional data (“difference” rather than “change”)

Methods

Please ensure all methods are written in past tense.

There are grammatical errors in sentence structure which will require editorial attention.

Re the statement : “Forty-two households still need to provide written consent.” Please clarify that these participants have therefore been excluded from this analysis.

How were day time naps recorded/dealt with?

Before creating ilrs, you need to check if there are any zero values in any of the compositional parts, or any NA values. All zeros need to be replaced with small values and all NA values must be removed before applying the ilr transformation. Please specify if/how many zeros were replaced in your methods.

Please clarify this sentence: “Applied the isometric log-ratio (ILR) transformation to compositional data, converting each composition (e.g., duration of SP, SB, LPA, and MVPA) into a logarithmic ratio relative to the geometric mean of other compositions [28].” To “: “An isometric log-ratio (ILR) transformation was applied [28].” It does not matter what type of ilr transformation is used for this model, so it is not necessary to describe it.

How did you account for potential clustering at the child care centre level?

Line 205: “d represents the number of compositions,” Please correct the terminology throughout this section. The “composition” refers to all the behaviours or compositional parts (sleep, sed, LPA and MVPA) together. You are using it here to refer to a single part. Due to multiple errors in this paragraph, I suggest omitting this technical/statistical paragraph and refer to the statistical papers instead. This is especially pertinent as you are using the compositional isotemporal substitution model for which the type of ilr transformation is irrelevant (you will get the same results regardless of the sequential binary partition used to make the ilrs).

Results:

“There is no significant difference between 224 arithmetic percentage and component percentage.” This was not tested by a statistical test, so please reword.

The pair-wise log-ratio presents co-dependency between behaviours, not “correlation” or “Association” – please reword the results around this.

Line 230 onwards – please also provide the multivariate test statistic for the overall composition. You may need to orient the table vertically to make it fit.

Line 245: “Based on the 95% confidence interval (CI), compositional isotemporal substitution was carried out for cognitive ability outcome measures.” Please remove the words “Based on the 95% confidence interval (CI),” so that this sentence makes sense.

Results and Discussion and Conclusion: throughout: when talking about reallocation results, please reword “increases” ,“decreases” and “changes” to “higher”, “lower” and “differences” to avoid causal/directional language.

Line 349: “The dose-response (Figures 1-3) result shows a significant asymmetry in pre-school children's total” please avoid the word “significant” as it implies a statistical test has been conducted to show that there is asymmetry.

Reviewer #2: This manuscript focuses on preschool children and explores the impact of 24-hour movement behaviors on cognitive ability, particularly the promoting effect of moderate to high intensity physical activity on cognitive ability. The study used compositional data analysis method to propose effective strategies for behavioral substitution, providing practical reference suggestions for preschool educators and parents. The manuscript provides a new perspective on exploring the cognitive development of preschool children by integrating different intensities of physical activity, sedentary behavior, and sleep.

This is a well-written manuscript that only needs to undergo a few minor changes:

1. The research involved 191 preschool children from Zhuzhou city in a cross-sectional study. ls this sample size sufficient to support the conclusions drawn in the study? And please ask the author to unify the sampling locations in the abstract and methods, so that readers can have a clearer understanding of the data analysis process in the manuscript and improve the readability of the article.

2. In the results section, SP, SB, LPA, and MVPA are almost all abbreviated throughout the manuscript, while in the descriptive statistics, the sleep period uses the full name. Please standardize the format for the author.

3. Please explain what the "-30-" in the title of the figure means and how it affects its readability. If necessary, please optimize the title of the figure.

4. Please improve your English expression.

Reviewer #3: Thank you for asking me to review this interesting and important article. The topic selection is valuable, but there are still some issues that need to be revised to further improve the quality of the manuscript.

Introduction

1. Line 55-56, “previous studies have found…”, citations are needed.

2. Line 59-60, this sentence is too wordy, please simplify it.

3. Line 73, citations are needed to support your idea.

Methods

1. In the methodology, there should be a flowchart of subject recruitment and eventual inclusion in the statistical analysis. Also, please report the number of boys and girls separately.

2. The sample size of this study is relatively small. Can it achieve sufficient statistical power? Please add the process of sample size estimation.

3. Line 124-126, citations are needed.

4. The activity characteristics of preschool children are short duration and high frequency, so the sampling interval is generally 1s. Why did this study use 15s?

5. The main exposure factor of this study is 24-hour movement behavior, including physical activity and sedentary time measured by objective methods, and sleep time measured by subjective methods. Usually the sum of the results measured by the two methods will not be exactly equal to 24 hours. How did the author process the data so that the time of physical activity + sedentary time + sleep time is equal to 24 hours?

6. How many examiners take part in the IQ test for preschool children? If there are multiple examiners, how can we ensure the consistency of the results from different examiners?

7. Line 170, citations are needed.

8. In addition to age and gender, other factors that affect preschool children's cognitive abilities should also be considered, such as BMI, parents' education, income, etc. In particular, maternal education level, previous research has shown that it has an extremely significant impact on preschool children's cognition.

9. Line 209, please cite references to support your explanation of isotemporal substitution model.

Discussions

1. Line 320-321, please express your idea in a scientific report way.

2. Why does the content of advantage appear before the title of “Strengths and limitations”?

6. PLOS authors have the option to publish the peer review history of their article (what does this mean? ). If published, this will include your full peer review and any attached files.

**Do you want your identity to be public for this peer review?** For information about this choice, including consent withdrawal, please see our Privacy Policy .

Reviewer #1: No

Reviewer #2: No

Reviewer #3: No

---

## [Author Response · Author response to Decision Letter 1]

4 Jan 2025

Response Letter

Dear editors and reviewers,

We are very grateful for your constructive comments and suggestions for our manuscript entitled “24-hour movement behaviors and cognitive ability in preschool children: A compositional and isotemporal reallocation analysis” (ID: PONE-D-24-28927). Your comments are very valuable and helpful for improving our manuscript. In the following, the responses to all the comments are provided one by one.

We have tried our best to make all the revisions clear, and we hope that the revised manuscript can satisfy the requirements for publication.

The main revisions in the new manuscript are:

1. The relevant literature on the impact of different sleep parameters and types of prolonged sitting on cognitive development in preschool children has been added.

2. The estimation of minimum sample size and recruitment flowchart have been updated.

3. The errors in the statistical paragraph have been corrected and relevant supporting references have been added.

4. The methods of sleep measurement and cognitive measurement were explained.

Sincerely,

Corresponding author.

Response to the comments of Co-EIC and Associate Editor

Co-EIC: When submitting your revision, we need you to address these additional requirements.

Q1. Please ensure that your manuscript meets PLOS ONE's style requirements, including those for file naming.

Response from the author: Thank you for your kind suggestion. The style of the revised manuscript and the naming of the files have been changed according to the requirements of PLOS ONE.

Q2. Please note that PLOS ONE has specific guidelines on code sharing for submissions in which author-generated code underpins the findings in the manuscript. In these cases, we expect all author-generated code to be made available without restrictions upon publication of the work.

Response from the author: Thank you very much for the editor's attention and recognition of our research work. We fully understand your journal's requirements for code sharing. Due to our laboratory's policies or confidentiality agreements, research codes are not publicly available. The corresponding code can be provided to qualified researchers upon request by the corresponding author.

Q3. Please note that funding information should not appear in any section or other areas of your manuscript. We will only publish funding information present in the Funding Statement section of the online submission form. Please remove any funding-related text from the manuscript.

Response from the author: Thank you for your kind suggestion. In the revised manuscript, the funding-related text has been removed from the manuscript.

Q4. Thank you for stating the following financial disclosure: “Hunan Provincial Education Science Planning Project Base Project, grant number XJK23AJD055”.

Response from the author: Funders played an important role in article quality monitoring, publication decisions, manuscript review, and editing.

Q5. We note that you have indicated that there are restrictions to data sharing for this study. PLOS only allows data to be available upon request if there are legal or ethical restrictions on sharing data publicly. 

Response from the author: Thank you very much for the editor's attention and recognition of our research work. We fully understand your journal's requirements for data sharing. Due to our laboratory's policies or confidentiality agreements, data supporting research results are not publicly available. The corresponding code can be provided to qualified researchers upon request by the corresponding author.

Data owner organization:

Organization Name: Hunan Research Centre for Excellence in Fitness

Email: wangshiqiang@hut.edu.cn

Q6. Your ethics statement should only appear in the Methods section of your manuscript. If your ethics statement is written in any section besides the Methods, please move it to the Methods section and delete it from any other section. Please ensure that your ethics statement is included in your manuscript, as the ethics statement entered into the online submission form will not be published alongside your manuscript.

Response from the author: Thank you very much for the constructive feedback. The ethical statement in this article has been removed from the section outside the methods and added to the methods section.

Response to the comments of Reviewer #1

Q1. Please rephrase “This is due to the subpar quality of the studies and the limited accumulation of research evidence.” Or provide evidence for why the studies are “subpar” – what does this mean?

Response from the author: The authors apologize for confusing the reviewer. The author wants to express that due to different existing research schemes, it is still unclear which PA intensity is more beneficial for cognitive ability.

The author has removed the original wording and rephrased “due to differences between the PA protocol and the results reported in previous research reports, it is currently unclear which elements of PA are more beneficial for the cognitive abilities of preschool children”.

Q2. For the statement “Moreover, different sleep parameters also have differences in the development of cognitive abilities in early children [11]” please provide some context of what these sleep parameters are. Will they be explored in this study?

Response from the author: As suggested by the reviewer, the authors have researched and added more literatures to support the differential effects of different sleep parameters on cognitive development in preschool children, and they are referred as [7] and [17] in the revised manuscript, as seen in line 58-60. In addition, this study only measured the sleep duration of preschool children. Therefore, this study only discusses the impact of sleep duration on cognitive development in preschool children.

Q3. “Similarly, the association between SB and cognitive development in early children should also vary depending on the type of sedentary behavior [12].” Can you please provide some examples here.

Response from the author: As suggested by the reviewer, the authors have researched and added more literatures to support the differential effects of different SB types on cognitive development in preschool children, and they are referred as [9] and [18] in the revised manuscript, as seen in line 64-54.

Q4. “The advantage of this analysis method is that it fully corresponds to the requirements of the behavioral guidelines, with satisfaction marked as 1 and dissatisfaction marked as 0, resulting in a clear and intuitive presentation of the data.” This is confusing to me. Does satisfaction mean “compliance with the guidelines” and how does this work when there are multiple guidelines each for different behaviours within the 24 h day, i.e., it’s possible to meet some guidelines (sleep and PA), and not meet others (screen time).

Response from the author: The authors apologize for confusing the reviewer. The author intends to highlight the advantages of using component data multiple linear regression by examining the shortcomings of the multiple mixed linear regression research method. Therefore, the author changed the originally unclear wording to address the shortcomings of the multiple linear regression research method. as seen in line 82-87.

Q5. “A study from low-income areas in Brazil found a positive correlation between 24-hour movement behaviors and EF in pre-school children” This doesn’t make sense – it’s not possible to have a positive correlation between all 24-h MBs and an outcome? Do you mean there was an association between the composition and EF?

Response from the author: The authors are sorry for our careless mistakes. What the author wants to express is that there is a correlation between 24-hour MBs and EF, and the positive has been removed, as seen in line 92-94.

Q6. Please refine the Aims

Aim 1: i think this is to evaluate the associations between the overall 24-h MB composition and outcomes?

Response from the author: Thank you for your kind suggestion. This is the most core content explored in this study. Due to the author's wording, it caused confusion for the reviewers. The authors deeply apologize and have made changes in the manuscript.

Q7. Aim 2: please reword to avoid inference of directionality which is not possible with cross-sectional data (“difference” rather than “change”)

Response from the author: Thank you for your insightful suggestion. Isotemporal Substitution Paradigm is based on the correlation between the proportion of existing 24-hour movement behavior time and the cognitive ability of preschool children, simulating two movement behaviors to replace each other at the same time, and predicting the expected theoretical changes in cognitive ability of preschool children. Therefore, the authors believe that changes are reasonable.

Q8. Please ensure all methods are written in past tense.

Response from the author: Thank you for your insightful suggestion. The authors have changed all the grammar in the research methodology section to the past tense.

Q9. There are grammatical errors in sentence structure which will require editorial attention.

Response from the author: Thank you for your insightful suggestion. A careful proof-reading has been conducted for the revised manuscript to improve the language and grammar.

Q10. Re the statement : “Forty-two households still need to provide written consent.” Please clarify that these participants have therefore been excluded from this analysis.

Response from the author: Thank you very much for the constructive feedback. 42 invited participants were excluded from this analysis due to their guardians not providing written consent.

Q11. How were day time naps recorded/dealt with?

Response from the author: Thank you for your insightful question. Before the start of this study, the researchers established WeChat groups with parents and teachers. The day time nap on weekdays is recorded by the teacher. However, it is relatively difficult for the teacher to record the bedtime and wake-up times of multiple children during their time in the kindergarten. Therefore, the teacher took photos of each child bedtime and wake-up times during the day and checked in the group. Researchers recorded the nap time of young children during the day by uploading photos from teachers. Parents record the nap time of young children during weekends and daytime, as seen in line 160-162.

Q12. Before creating ilrs, you need to check if there are any zero values in any of the compositional parts, or any NA values. All zeros need to be replaced with small values and all NA values must be removed before applying the ilr transformation. Please specify if/how many zeros were replaced in your methods.

Response from the author: Thank you for your insightful question. During the data collection process, there were 27 preschool children with missing 24-hour movement behavior data. Therefore, the data of 27 preschool children has been excluded before applying the ilr transformation.

Q13. Please clarify this sentence: “Applied the isometric log-ratio (ILR) transformation to compositional data, converting each composition (e.g., duration of SP, SB, LPA, and MVPA) into a logarithmic ratio relative to the geometric mean of other compositions [28].” To “: “An isometric log-ratio (ILR) transformation was applied [28].” It does not matter what type of ilr transformation is used for this model, so it is not necessary to describe it.

Response from the author: Thank you for your insightful suggestion. As suggested by the reviewer, the author simplified "Applied the isometric log ratio (ILR) transformation to compositional data, converting each composition (e.g. duration of SP, SB, LPA, and MVPA) into a logarithmic ratio relative to the geometric mean of other compositions [28]" and simplified it to "An isometric log ratio (ILR) transformation was applied [28]", as seen in line 234.

Q14. How did you account for potential clustering at the child care centre level?

Response from the author: Thank you for your insightful question. The preschool education in the urban area of Zhuzhou City is divided into four sections. This study strategically selected one kindergarten from each of the four districts, in order to ensure a representative sample among different education departments in the city, as seen in line 118-119.

Q15. Line 205: “d represents the number of compositions,” Please correct the terminology throughout this section. The “composition” refers to all the behaviours or compositional parts (sleep, sed, LPA and MVPA) together. You are using it here to refer to a single part. Due to multiple errors in this paragraph, I suggest omitting this technical/statistical paragraph and refer to the statistical papers instead. This is especially pertinent as you are using the compositional isotemporal substitution model for which the type of ilr transformation is irrelevant (you will get the same results regardless of the sequential binary partition used to make the ilrs).

Response from the author: Thank you for your insightful suggestion. As suggested by the reviewer, the authors omitted this statistical paragraph and referred to relevant statistical papers, as seen in line 245.

Q16. “There is no significant difference between 224 arithmetic percentage and component percentage.” This was not tested by a statistical test, so please reword.

Response from the author: Thank you for your insightful suggestion. The authors omitted this sentence.

Q17. The pair-wise log-ratio presents co-dependency between behaviours, not “correlation” or “Association” – please reword the results around this.

Response from the author: Thank you very much for the constructive feedback. The author changed "correlation" or "association" to "co-dependent", as seen in line 265-267.

Q18. Line 230 onwards – please also provide the multivariate test statistic for the overall composition. You may need to orient the table vertically to make it fit.

Response from the author: Thank you very much for the constructive feedback. The multivariate test statistics for the overall composition are shown in Table 3, as seen in page 13.

Q19. Line 245: “Based on the 95% confidence interval (CI), compositional isotemporal substitution was carried out for cognitive ability outcome measures.” Please remove the words “Based on the 95% confidence interval (CI),” so that this sentence makes sense.

Response from the author: Thank you very much for the constructive feedback. As suggested by the reviewer, the authors have deleted “Based on the 95% confidence interval (CI)”.

Q20. Results and Discussion and Conclusion: throughout: when talking about reallocation results, please reword “increases” ,“decreases” and “changes” to “higher”, “lower” and “differences” to avoid causal/directional language.

Response from the author: Thank you very much for the constructive feedback. When reallocating the results, it is to compare the expected theoretical results with the results of the study on the impact of the proportion of 24-hour movement behavior time on the cognitive ability of preschool children. Therefore, the authors believe that the use of words such as "increases," "decreases," and "changes" are reasonable.

Q21. “The dose-response (Figures 1-3) result shows a significant asymmetry in pre-school children's total” please avoid the word “significant” as it implies a statistical test has been conducted to show that there is asymmetry.

Response from the author: Thank you very much for the constructive feedback. The authors removed the word “significant” and changed “The dose-response (Figs 1-3) result shows a significant asymmetry in pre-school children's total” to “The dose-response (Figs 1-3) results revealed that there was asymmetry in the changes of FIQ in preschool children when MVPA and other movement behaviors were

---

## [Decision Letter · Decision Letter 1]

16 Jan 2025

PONE-D-24-28927R1学龄前儿童 24 h 运动行为与认知能力的组合和等时空重新分配分析PLOS ONE

Dear Dr. Wang,

Thank you for submitting your manuscript to PLOS ONE. After careful consideration, we feel that it has merit but does not fully meet PLOS ONE’s publication criteria as it currently stands. Therefore, we invite you to submit a revised version of the manuscript that addresses the points raised during the review process.

**Please pay attention to the concerns of the reviewers and address their worries**

We look forward to receiving your revised manuscript.

Kind regards,

Jindong Chang, Ph.D.

Academic Editor

PLOS ONE

**Journal Requirements:**

Reviewers' comments:

Reviewer's Responses to Questions

**Comments to the Author**

1. If the authors have adequately addressed your comments raised in a previous round of review and you feel that this manuscript is now acceptable for publication, you may indicate that here to bypass the “Comments to the Author” section, enter your conflict of interest statement in the “Confidential to Editor” section, and submit your "Accept" recommendation.

Reviewer #2: (No Response)

Reviewer #3: (No Response)

2. Is the manuscript technically sound, and do the data support the conclusions?

Reviewer #2: Yes

Reviewer #3: Yes

3. Has the statistical analysis been performed appropriately and rigorously? 

Reviewer #2: Yes

Reviewer #3: Yes

4. Have the authors made all data underlying the findings in their manuscript fully available?

Reviewer #2: Yes

Reviewer #3: Yes

5. Is the manuscript presented in an intelligible fashion and written in standard English?

Reviewer #2: Yes

Reviewer #3: Yes

6. Review Comments to the Author

**Reviewer #2:**  The manuscript has made significant improvements, but there are two minor comments.

Firstly, standardize the format of accelerometer name, as seen in line 139-140.

Secondly, the manuscript title displayed in the submission system is in Chinese. Please modify it.

**Reviewer #3:**  Most of the comments have been addressed in the revised manuscript, but there is one minor comment.

This study aims to explore the relationship between 24-hour movement behaviors and cognitive ability, not to estimate the prevalence of a certain disease. Therefore, is the prevalence parameter applicable when estimating the sample size? You can refer to the following information https://sample-size.net/correlation-sample-size/

Other suggestions

1. The title displayed in the submission system is in Chinese. Please modify it.

2. In the revised content, all the texts are piled together. There is not even a blank line to distinguish the replies to different questions or the replies to different reviewers. It is extremely difficult to read, and I hope it can be improved in the future.

7. PLOS authors have the option to publish the peer review history of their article (what does this mean? ). If published, this will include your full peer review and any attached files.

**Do you want your identity to be public for this peer review?** For information about this choice, including consent withdrawal, please see our Privacy Policy .

Reviewer #2: No

Reviewer #3: No

---

## [Author Response · Author response to Decision Letter 2]

8 Feb 2025

Dear editors and reviewers,

We are very grateful for your constructive comments and suggestions for our manuscript entitled “24-hour movement behaviors and cognitive ability in preschool children: A compositional and isotemporal reallocation analysis” (ID: PONE-D-24-28927). Your comments are very valuable and helpful for improving our manuscript. In the following, the responses to all the comments are provided one by one.

We have tried our best to make all the revisions clear, and we hope that the revised manuscript can satisfy the requirements for publication.

The main revisions in the new manuscript are:

1. Revised the formatting issues in the previous manuscript.

2. To ensure the completeness of Table 4, its position has been adjusted.

3. Unified the names of accelerometers.

4. Improved the process of calculating appropriate sample size.

Sincerely,

Corresponding author.

Response to the comments of Reviewer #2

Q1. Standardize the format of accelerometer name, as seen in line 139-140.

Response from the author: The authors are sorry for our careless mistakes. As suggested by the reviewer, accelerometer name has been unified in the revised manuscript, as seen in line 141-142.

Q2. The manuscript title displayed in the submission system is in Chinese. Please modify it.

Response from the author: The authors are sorry for our careless mistakes. As suggested by the reviewer, the manuscript title has been changed to English in the submission system.

Response to the comments of Reviewer #3

Q1. This study aims to explore the relationship between 24-hour movement behaviors and cognitive ability, not to estimate the prevalence of a certain disease. Therefore, is the prevalence parameter applicable when estimating the sample size?

Response from the author: Thank you very much for the constructive feedback. This study adopted a reasonable and appropriate sample size calculation method. The previous manuscript imitated Mota et al.'s writing paradigm due to the lack of a good language expression. The authors attempted to find a more reasonable explanation for the calculation process of appropriate sample size based on relevant references of G*power's appropriate sample size calculation method, as seen in line 126-130.

Q2. The manuscript title displayed in the submission system is in Chinese. Please modify it.

Response from the author: The authors are sorry for our careless mistakes. As suggested by the reviewer, the manuscript title has been changed to English in the submission system.

Q3. In the revised content, all the texts are piled together. There is not even a blank line to distinguish the replies to different questions or the replies to different reviewers. It is extremely difficult to read, and I hope it can be improved in the future.

Response from the author: Thank you very much for the constructive feedback. We are so sorry that the authors overlooked the reviewers' reading experience of the revised manuscript in order to try their best to revise the reviewers' comments. We will pay attention in the future.

---

## [Editor Report · Decision Letter 2]

13 Feb 2025

24-hour movement behaviors and cognitive ability in preschool children: A compositional and isotemporal reallocation analysis

PONE-D-24-28927R2

Dear Dr. Wang,

We’re pleased to inform you that your manuscript has been judged scientifically suitable for publication and will be formally accepted for publication once it meets all outstanding technical requirements.

Kind regards,

Jindong Chang, Ph.D.

Academic Editor

PLOS ONE
---

## [Editor Report · Acceptance letter]

PONE-D-24-28927R2

PLOS ONE

Dear Dr. Wang,

I'm pleased to inform you that your manuscript has been deemed suitable for publication in PLOS ONE. Congratulations! Your manuscript is now being handed over to our production team.

Kind regards,

on behalf of

Dr. Jindong Chang

Academic Editor

PLOS ONE